# Association between the Temporomandibular Joint Morphology and Chewing Pattern

**DOI:** 10.3390/diagnostics13132177

**Published:** 2023-06-26

**Authors:** Sasin Sritara, Yoshiro Matsumoto, Yixin Lou, Jia Qi, Jun Aida, Takashi Ono

**Affiliations:** 1Department of Orthodontic Science, Graduate School of Medical and Dental Sciences, Tokyo Medical and Dental University (TMDU), Tokyo 113-8510, Japan; sasin.sri@mahidol.edu (S.S.);; 2Department of Orthodontics, Faculty of Dentistry, Mahidol University, Bangkok 10400, Thailand; 3Department of Oral Health Promotion, Graduate School of Medical and Dental Sciences, Tokyo Medical and Dental University (TMDU), Tokyo 113-8510, Japan; aida.ohp@tmd.ac.jp

**Keywords:** temporomandibular joint, mandibular condyle, masticatory function, cone-beam computed tomography, multiple linear regression analysis

## Abstract

This study aimed to investigate whether the morphology of the temporomandibular joint (TMJ) is associated with chewing patterns while considering skeletal morphology, sex, age, and symptoms of temporomandibular disorder (TMD). A cross-sectional observational study of 102 TMJs of 80 patients (age 16–40 years) was performed using pretreatment records of cone-beam computed tomography imaging of the TMJ, mandibular kinesiographic records of gum chewing, lateral and posteroanterior cephalometric radiographs, patient history, and pretreatment questionnaires. To select appropriate TMJ measurements, linear regression analyses were performed using TMJ measurements as dependent variables and chewing patterns as the independent variable with adjustment for other covariates, including Nasion-B plane (SNB) angle, Frankfort-mandibular plane angle (FMA), amount of lateral mandibular shift, sex, age, and symptoms of TMD. In multiple linear regression models adjusted for other covariates, the length of the horizontal short axis of the condyle and radius of the condyle at 135° from the medial pole were significantly (*p* < 0.05) associated with the chewing patterns in the frontal plane on the working side. “Non-bilateral grinding” displayed a more rounded shape of the mandibular condyle. Conversely, “bilateral grinding” exhibited a flatter shape in the anteroposterior aspect. These findings suggest that the mandibular condyle morphology might be related to skeletal and masticatory function, including chewing patterns.

## 1. Introduction

The temporomandibular joint (TMJ) is the most unique joint in the body, as the two joints must move in synchronism. The TMJ can develop morphological bone remodeling as a result of various factors such as masticatory activities, aging, temporomandibular disorder (TMD), and occlusal changes [1,2,3,4]. Previous studies have found that the size and shape of the TMJ are associated with skeletal morphology, such as the size of the mandible and vertical growth patterns [5]. The concept that the TMJ is a component of an articular triad, with the TMJ providing two points of contact and dentition providing the third, regulating mandibular movement, has been widely broadcast. Consequently, changes in mandibular movement may be linked to alterations in the structure of the TMJs and occlusion [6].

Previous studies have demonstrated that different skeletal patterns show significant differences in condylar morphology, joint space, joint-fossa morphology, and condylar position [7,8]. Patients with a larger Frankfort-mandibular plane angle (FMA) had a smaller linear measurement of the mandibular condyle size, although the difference was not statistically significant [5,9]. Mendoza et al. [5] and Nakawaki et al. [10] discovered that there were significant differences between hyperdivergent and hypodivergent patients, with the hypodivergent group presenting larger volumes of the mandibular condyle.

Lateral mandibular shift, also known as facial asymmetry, is a common craniofacial deformity, defined by a lateral deviation of the midline of the mandible caused by asymmetric growth. Previous studies found that the structures of the TMJ on the nondeviated side of the jaw were larger than those on the deviated side in patients with mandibular asymmetry due to excessive growth on the nondeviated side [11,12].

Masticatory activity is a complex motor activity that is primarily controlled by the central pattern generator of the masticatory system. The central pattern generator receives sensory input from various stomatognathic organs and neuromuscular systems to determine chewing stroke. The tear-shaped pattern in the frontal plane describes a typical chewing stroke [6]. The influence of dental occlusion on chewing patterns has long been a topic of study in the field of dentistry. Many researchers have studied the relationship between chewing movements and the occlusion of teeth. Ahlgren was the first to establish chewing patterns in the frontal plane using direct observation. However, after evaluating the chewing patterns of 290 patients with malocclusion and 30 patients with normal occlusion, no relationship between chewing patterns and malocclusion type was discovered [13]. A mandibular kinesiograph was then used to validate the mandibular movements. According to Nie et al. [14], patients with posterior crossbite may have more abnormal chewing types than those with an anterior crossbite. Moreover, posterior crossbite may contribute to the high frequency of reverse and crossover chewing types, particularly when accompanied by lateral mandibular shift, mandibular prognathism, or arch crossbite. However, in a study of 247 patients, Proeschel discovered that some participants with normal occlusion and some who had already undergone orthognathic surgery still displayed abnormal chewing patterns [15]. These findings suggest that other key elements affect chewing movement. Previous studies have identified different characteristic pattern distributions of normal occlusion and malocclusion rather than a single chewing pattern, and a significant proportion of normal chewing strokes were observed in the normal occlusion group. Malocclusion is more common in chopping, reversed, and crossover chewing strokes [13,14,15]. Kim et al. classified two typical chewing patterns [16]; one is more vertical in character, resembling a chopping movement, while the other has a more flattened lateral guidance, also known as “grinding movement.”

Recent studies have reported an association between flattening of the lateral guidance in the frontal plane of chewing patterns, steepening of the condylar pathway, and deep glenoid fossa with steep eminence [17,18]. The morphology of the TMJ is influenced by numerous factors, although cause-and-effect relationships are mostly unknown. Currently, little is known about the relationship between TMJ morphology and chewing patterns. We hypothesized that the impact of masticatory activity on the condyle fossa relationship is inevitable and that significant associations exist between the morphology of the TMJ and different chewing patterns. This study aimed to investigate whether TMJ morphology is associated with chewing patterns as well as skeletal morphology, sex, age, and symptoms of TMD.

## 2. Materials and Methods

### 2.1. Study Design and Ethics

This study was designed as a retrospective cross-sectional observational study and was approved by the Ethics Committee of Tokyo Medical and Dental University Hospital (approval #: D2021-073). The study population comprised eighty patients aged 16–40 years who received orthodontic treatment at Tokyo Medical and Dental University Hospital between May 2013 and January 2020 and needed cone-beam computed tomography (CBCT) imaging (3DX Multi-Image Micro CT FPD8, Morita Co., Ltd., Kyoto, Japan) of the TMJ for various clinical reasons and had records of mandibular movement at the pretreatment examination.

In this study, we examined the association between TMJ morphology and bilateral chewing patterns. The independent variable, bilateral chewing patterns, and the dependent variable, the morphology of the TMJ, were adjusted for skeletal morphology, sex, age, and symptoms of TMD as covariates in the multiple linear regression models.

An a priori power analysis showed that at a 95% confidence level, a sample size of 103 would provide an 80% probability of demonstrating a medium effect (f = 0.15) for the association between TMJ morphology and chewing patterns using a multiple linear regression model after adjusting for seven independent variables in total. Patients with craniofacial anomalies, major head and neck injuries, or a history of head and neck surgery were excluded from the study. According to our TMJ side selection criteria, 58 TMJs on the deviated side of patients with mandibular shift and 22 on the left, and 22 on the right side of those without mandibular shift were selected. In total, 102 TMJs were used in this study.

### 2.2. Measurements of TMJ

CBCT was used to examine the three-dimensional morphology of the TMJ. The patients were seated in a posture in which the Frankfort horizontal plane was horizontal and the jaws were in centric occlusion when CBCT images were taken. For image reconstruction, a single 360° scan was used to collect projection data from a cylinder (height, 30 mm; diameter, 40 mm); 80 kV, 7 mA, and 17 s. The reconstructed slices were 1 mm thick. TMJ morphology was assessed using three-dimensional analysis software (AW server 3.2 Ext 4.0, GE Healthcare, Chicago, IL, USA). The measurements of TMJ were performed in the horizontal, coronal, and sagittal planes, as shown in Figure 1, which was modified from previous studies [2,7,19]. After constructing the three-dimensional rendered image, the measurement of the TMJ started by scrolling through the horizontal slices and then selecting the slice with the long axis of the condyle in the horizontal plane (HLC). Then, the short axis of the condyle in the horizontal plane (HSC) perpendicular to the HLC and the horizontal condylar angle (HCA), which was measured from a reference line perpendicular to the midsagittal plane, was measured (Figure 1A). Subsequently, the slice in the center of the mandibular condyle was selected as coronal slices parallel to the HLC and sagittal slices perpendicular to the HLC. In the selected coronal slice, we measured the long axis of the condyle in the coronal plane from the medial pole to the lateral pole, the radius of the condyle at 45°, 90° (C90), and 135° from the medial pole (C135), joint space at 45°, joint space at 90°, and the joint space at 135° (Figure 1B). Finally, we measured the height of the condyle in the sagittal plane (SHC), the depth of the condyle in the sagittal plane (SDC), anterior joint space, posterior joint space, and articular eminence inclination from the selected sagittal slice (Figure 1D). To evaluate intraobserver reliability, the same observer repeated all measurements after six weeks, blinded to the previous measurements. A different observer evaluated the left and right TMJs of 30 patients and compared the results to the first observer’s measurement to evaluate interobserver reliability.

### 2.3. Chewing Movement

The pretreatment path of the mandibular incisors of the patients was recorded using a mandibular kinesiograph (MKG) system (K7-I craniomandibular evaluation system, Myotronics-Noro Med, Seattle, WA, USA). MKG recordings of gum chewing on the left and right sides in the frontal plane were utilized using a modified method described in previous studies to determine the main chewing patterns as grinding, chopping, reversed, or crossover, as shown in Figure 2 [14,15].

Those who showed grinding patterns on both sides were defined as the “Bilateral grinding” group, whereas those who showed chopping, reversing, or crossing patterns on at least one side were defined as the “Non-bilateral grinding” group, as shown in Figure 3.

### 2.4. Skeletal Morphology, Age, Sex, and Symptoms of TMD

Lateral and posteroanterior (PA) cephalometric radiographs were obtained using a cephalostat (Axiom Aristos VX, Siemens, Munich, Germany). Tracings of lateral cephalograms were used to evaluate the angle between the Sella–Nasion plane and the Nasion-B plane (the SNB angle), which assesses the anteroposterior position of the mandible relative to the upper cranial structures and the FMA (Figure 4A). The landmarks of the PA cephalogram were identified using the methods proposed by Sassouni to evaluate the lateral mandibular shift (Figure 4B) [20].

Data on age, sex, and history of symptoms of TMD were collected from the patients’ histories and pretreatment questionnaires. The sex was listed as either male or female, and the age of each individual was determined using the number of complete years. The history of TMD included sounds, pain, and trismus. Patients were categorized as having or not having a history of TMD.

### 2.5. Statistical Analysis

Statistical analysis was based on the first measurement of the TMJ. The collected data were coded and analyzed using STATA software (Stata Statistical Software version 17.0, Stata Corp LP, College Station, TX, USA). The intra- and interobserver reliabilities of all TMJ measurement parameters were evaluated using a one-sample t-test. Univariate analysis was conducted between different measurements of the TMJ and chewing patterns to identify variables (*p* <  0.2) [21], which were included in the multiple linear regression analysis. In the multiple linear regression models, the dependent variable was TMJ morphology, whereas the independent variables were bilateral chewing patterns, the SNB angle, the FMA angle, the amount of lateral mandibular shift, sex, age, and symptoms of TMD. The variables whose *p*-value was less than 0.05, from multiple linear regressions, were declared as statistically significant.

## 3. Results

### 3.1. Three-Dimensional Reconstruction of TMJ

The CBCT images of TMJ from groups “Bilateral grinding” and “Non-bilateral grinding” were three-dimensionally reconstructed and observed as shown in Figure 5A,B.

### 3.2. Statistical Analysis

Based on the criteria we established for selecting TMJ sides, a total of 58 TMJs located on the deviated side were chosen from patients exhibiting mandibular shift. Additionally, 22 TMJs were selected from the left side and 22 from the right side of individuals without mandibular shift. In total, the study utilized 102 TMJs to gather data and conduct analyses. The total sample size was 102 TMJs from 80 patients; the characteristics of each variable used in this study are listed in Table 1. All measurements of the TMJ showed no statistically significant difference between the intra- and interobserver reliabilities at *p* > 0.05, as shown in Table 2.

From the univariate analysis (Table 3), HSC, HCA, C90, C135, SHC, and SDC were the measurement items of TMJ that were eligible for the multiple linear regression models (*p* < 0.2). The details of each multiple linear regression model are listed in Table 4. The *p*-values of chewing patterns in each of the multiple linear regression models, including the averages and standard deviations of the TMJ measurements, are shown in Table 5.

There was a significant association among chewing patterns, amount of lateral mandibular shift, sex, age, and the FMA angle on HSC and C135 (*p* < 0.05) in multiple linear regression models adjusted for covariates. HSC was positively associated with the chewing pattern group “Non-bilateral grinding”. Accordingly, when adjusted with the same covariates, C135 also showed a significant positive association with the chewing pattern group “Non-bilateral grinding” (*p* < 0.05). The association between the chewing patterns group and SDC was *p* = 0.066. The results of SDC with the chewing patterns group were not statistically significant, although they did coincide with the results of HSC and C135 (Figure 6).

According to the multiple linear regression models, the size of the mandibular condyle was negatively associated with the FMA angle, the amount of lateral mandibular shift, and the female sex. The vertical dimensions of the mandibular condyle (C90, C135, and SHC) were positively associated with the SNB angle (Table 4).

## 4. Discussion

To the best of our knowledge, this is the first study to demonstrate a significant association between TMJ morphology and bilateral chewing patterns in multiple linear regression models adjusted for the SNB angle, the FMA angle, amount of lateral mandibular shift, sex, age, and symptoms of TMD as covariates. The anteroposterior aspects of the mandibular condyle were the HSC and SDC, whereas the superolateral aspect was described by C135. The lengths of HSC and C135 were positively associated with the chewing pattern group “Non-bilateral grinding” when compared to the chewing pattern group “Bilateral grinding”. Although the results of the SDC multiple linear regression model with bilateral chewing patterns and other covariates were not statistically significant (*p* = 0.066), they did correspond with those of HSC and C135 (Figure 6). Figure 6 describes the relationship between the anteroposterior aspect of the mandibular condyle (HSC and SDC) and chewing patterns compared with the mediolateral aspect (HLC). The differences in the anteroposterior aspects were much greater. From the horizontal slice, the chewing pattern of the group “Non-bilateral grinding” presented a much rounder shape of the mandibular condyle, while the chewing pattern of the group “Bilateral grinding” presented a flatter shape in the anteroposterior aspect and the sagittal slice; the chewing pattern of the group “Non-bilateral grinding” exhibited a larger SDC compared to the chewing pattern of the group “Bilateral grinding”. Multiple linear regression models demonstrated significant correlations between specific TMJ morphology measurement items and chewing patterns (*p* < 0.05). Furthermore, in the present study, significant correlations were reported between the size of the mandibular condyle and the SNB angle, the FMA angle, the amount of lateral mandibular shift, and sex.

The previous studies suggested that the structures of the TMJ on the nondeviated side of the jaw were larger than those on the deviated side in patients with lateral mandibular shifts [11,12]. The deviated side of the TMJ was selected in our study and adjusted with the amount of lateral mandibular shift in the statistical model. Our findings are consistent with those of previous studies [11,12]. The nondeviated sides, which were larger than the deviated sides, were not included because this would cause the characteristics of the TMJ to become less distinctive. Preliminary studies revealed that most cases of large lateral mandibular deviation in our sample were skeletal Class III cases and that the condylar morphology of the nondeviated sides was large due to overgrowth. Therefore, we thought that including the mandibular condyles on the nondeviated sides would mask the morphological characteristics of the mandibular condyles on the deviated sides. One of the purposes of this study was to clarify the relationship between mandibular condyle morphology and gum chewing patterns on the deviated sides. Therefore, we decided not to include the mandibular condyles on the nondeviated sides in cases where mandibular deviation was evident.

The size of the TMJ on the deviated side was found to have a significant negative relationship with the amount of lateral mandibular shift (*p* < 0.05). The age range of the study population was confined to 16–40 years, and age was included as an independent variable in the multiple linear regression models. Correlations between morphological changes in the TMJ, sex, age, and symptoms of TMD are a major topic of discussion in various studies [3,4]. Sex, age, and symptoms of TMD are inevitably correlated independent variables that must be included in the statistical model. The prevalence of TMD was reported to be much higher in women, and symptoms of TMD have been reported to be related to age [4]. Thus, the coefficients of the independent variables may have been undermined.

According to Negishi et al. [22], chewing patterns can be altered through chewing exercises. Chewing gum was initiated 3 months after sagittal split ramus osteotomy and was performed for 5 min twice a day for 3 months. Following the exercise, the masticatory width increased significantly, suggesting a natural adaptation from narrow chewing patterns in the frontal plane to grinding chewing patterns as a result of morphological changes in the TMJ.

Patients with internal derangements of the TMJ had a significantly restricted range of chewing movement in both the lateral and vertical dimensions, as well as movement deceleration, compared to individuals without internal derangements of the TMJ [23,24]. Furthermore, different stages of internal derangement show varying degrees of aberrant chewing movement, according to Kuwahara et al. [23]: Patients with TMJ disc displacement without reduction were found to have the most chewing impairment, followed by TMJ disc displacement without reduction with perforation and TMJ disc displacement with a late reduction. However, no study has specified a standard criterion for diagnosing internal derangement of the TMJ.

Recent studies have shown that steepening of the condylar pathway or deepening of the articular eminence inclination is associated with the preferred chewing side, which has flatter lateral guidance or can be described as a grinding pattern compared with the opposite side [17,18,25]. We did not find a significant association between articular eminence inclination and chewing patterns in our study. The different results could be due to different focus points in bilateral chewing patterns or the preferred chewing side. According to our findings, an increase in the lateral width of the chewing pattern was significantly associated with a decrease in the anteroposterior and superolateral aspects of the mandibular condyle. This suggests that mandibular condyle remodeling may occur primarily in the anteroposterior and superolateral aspects, as the chewing patterns may shift to a grinding pattern with greater lateral width, either as a physiological adaptation or as a result of pathological bone resorption.

Balcioglu et al. [26] discovered that in patients with the preferred chewing side, the volumes of both the inferior and superior heads of the lateral pterygoid muscle on the affected side were significantly greater than those on the unaffected side. The lateral pterygoid muscle, which is essential for facilitating mastication and mandibular movement, is attached to the mandibular condyle. Recent anatomical investigations have focused on the insertion area of the lateral pterygoid muscle, not only the anterior aspect of the pterygoid fovea but also the medial aspect of the mandibular condyle [27,28]. While the present study’s findings suggest that mandibular condyle remodeling occurs largely in the anteroposterior and superolateral aspects, since the muscle force vector of the lateral pterygoid muscle in mandibular movements is defined by its origin and insertion point, this circumstance may affect the function of the lateral pterygoid muscle.

Another challenge is constructing multivariable statistical models and selecting independent variables. A careful and critical selection of variables is crucial to avoid multicollinearity among the independent variables [29]. The independent variables were chewing patterns and the dependent variable was the morphology of the TMJ while adjusting for the SNB angle, the FMA angle, the amount of lateral mandibular shift, sex, age, and symptoms of TMD as other covariates in the statistical models. These independent variables were biologically relevant factors that were known or based on a scientific hypothesis. The models included the SNB angle to represent the anteroposterior position of the mandible in relation to the upper cranial structures, and the FMA angle to represent the vertical growth pattern. Since the SNB angle, as one of the independent variables, has a high association with measurements related to dental occlusion such as Angle’s classification, overjet, and overbite, we did not include direct dental occlusion parameters in our statistical model. Table 2 presents the results of the analysis conducted to assess the intra- and interobserver reliabilities of the TMJ measurements. The findings reveal that there were no statistically significant differences observed between the levels of reliability for both intra- and interobserver assessments, as indicated by *p*-values greater than 0.05. This implies that the agreement or consistency among observers and the consistency over time for the same observer were comparable for all TMJ measurements evaluated in the study. Therefore, the data in Table 2 suggest that the TMJ variables examined exhibited similar levels of reliability regardless of whether they were assessed by the same observer on multiple occasions or by different observers independently. The findings of this study do not represent the general population because the samples were collected from individuals seeking orthodontic treatment. The study population and the estimated sample size from the power analysis were comparable.

The new findings demonstrated that there was a significant correlation between TMJ morphology and chewing patterns, even when considered with other covariates in multiple linear regression models, contributing to a better understanding of the TMJ in the fields of orthodontics and dentistry. Considering the previously reported association between occlusion and masticatory movement patterns [13,14,15,16], this study suggests that orthodontic treatment to improve occlusion may have some effect on TMJ morphology.

Further research is needed to clarify the possible mechanism, as well as whether this is a physiological adaptation, pathological condition, or merely a random coincidence [30]. Prospective studies with larger sample sizes and with an improvement in the study-population selection criteria are needed to clarify the cause-and-effect scenario of these correlations.

## 5. Conclusions

The findings of this study indicate that the anteroposterior and superolateral aspects of the mandibular condyle were significantly associated with chewing patterns in our statistical models. The findings of this study suggest that the morphology of the mandibular condyle may be associated with not only skeletal morphology but also masticatory function, such as chewing patterns that are related to occlusion and orthodontic treatment.

## Figures and Tables

**Figure 1 diagnostics-13-02177-f001:**
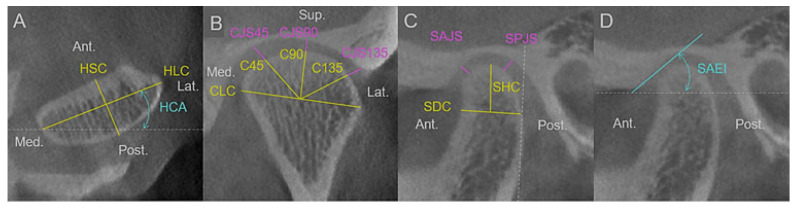
(**A**) Horizontal slice; HLC, long axis of condyle; HSC, short axis of condyle; HCA, horizontal condylar angle (**B**) coronal slice; CLC, long axis of condyle; C45, radius of condyle at 45°; C90, radius of condyle at 90°; C135, radius of condyle at 135°; CJS45, joint space at 45°; CJS90, joint space at 90°; CJS135, joint space at 135° (**C**,**D**) sagittal slice; SHC, height of condyle; SDC, depth of condyle; SAJS, anterior joint space; SPJS, posterior joint space; SAEI, articular eminence inclination. Ant., anterior; Post., posterior; Med., medial; Lat., lateral; Sup., superior.

**Figure 2 diagnostics-13-02177-f002:**
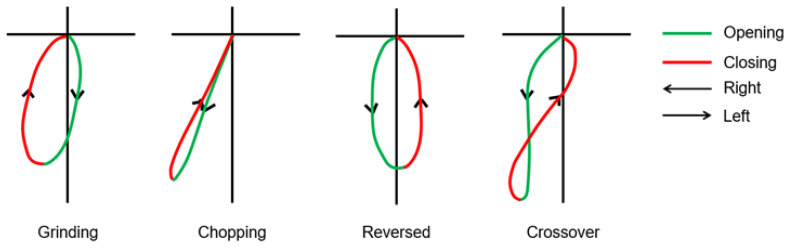
Grinding chewing patterns have a more lateral form than other chewing patterns, with an opening toward the nonworking side and a closing from the working side, and the opening and closing paths are wide apart in the frontal plane. Chopping, reversed, or crossover chewing patterns are classified as abnormal chewing patterns that have a mixed or ambiguous sense of direction and end with a form that has reversed sequencing.

**Figure 3 diagnostics-13-02177-f003:**
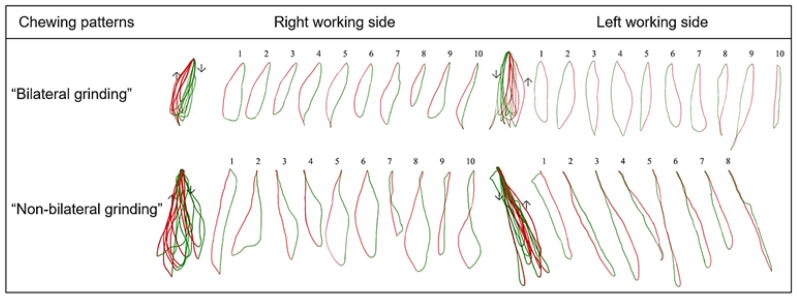
The main chewing patterns are determined as grinding, chopping, reversed, or crossover chewing patterns after each chewing stroke is evaluated. Chewing pattern group “Bilateral grinding”: grinding patterns on both sides. Chewing pattern group “Non-bilateral grinding”: with chopping, reversed, or crossover patterns on at least one side.

**Figure 4 diagnostics-13-02177-f004:**
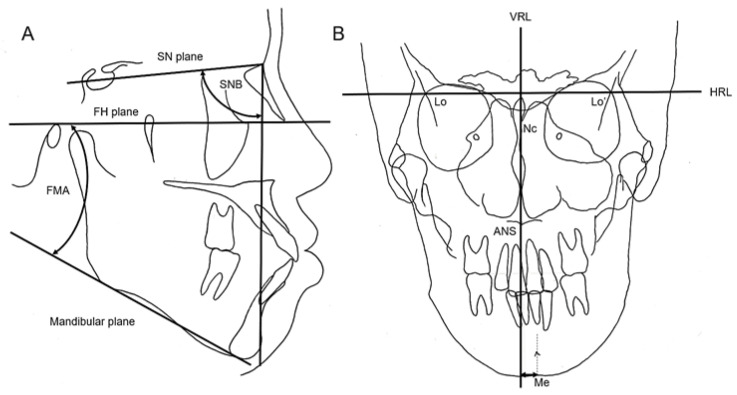
(**A**) The SNB angle and Frankfort-mandibular plane angle (the FMA angle) are measured from the lateral cephalogram tracing. The angle between the Sella-Nasion plane and the Nasion -B plane is known as the SNB angle. The intersection of the Frankfort horizontal plane and the mandibular plane forms the FMA angle. (**B**) The landmarks of the posteroanterior (PA) cephalogram are defined using Sassouni’s method for evaluating facial asymmetry and lateral mandibular shift. The vertical reference line (VRL) is drawn perpendicular to the horizontal reference line (HRL), passing through the neck of the crista galli (Nc), and connecting the bilateral intersection of the oblique orbital line with the lateral contour of the right-and left-side orbits (Lo-Lo’). The distance of the Menton (Me) from the VRL is used to assess the degree of lateral mandibular shift.

**Figure 5 diagnostics-13-02177-f005:**
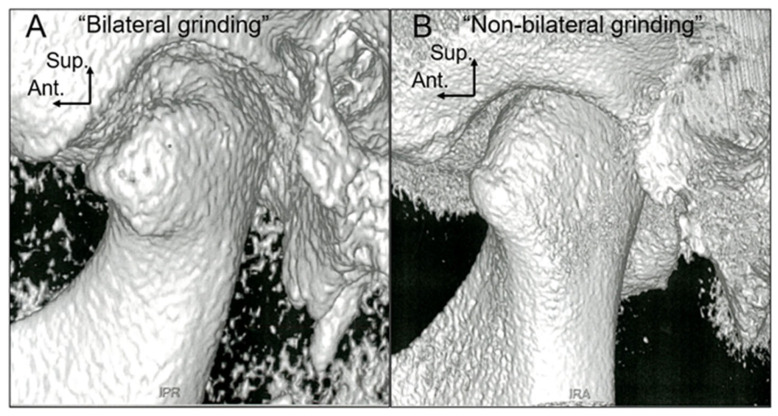
The three-dimensional reconstruction images of the TMJ cone-beam computed tomography of patients from chewing pattern group “Bilateral grinding” (**A**) and group “Non-bilateral grinding” (**B**).

**Figure 6 diagnostics-13-02177-f006:**
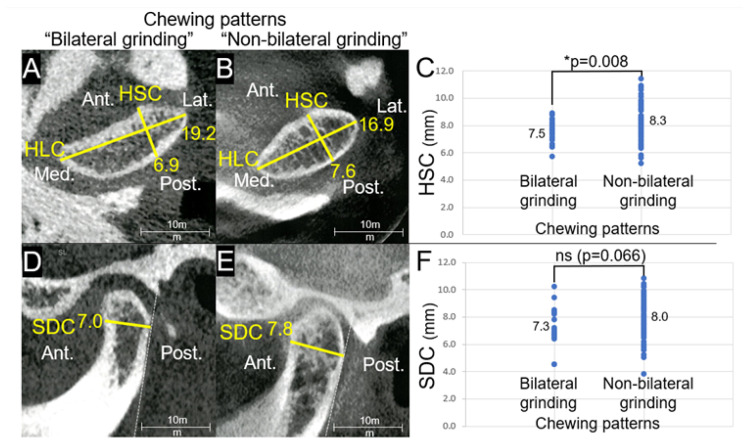
(**A**) Measurements of the horizontal short axis of the condyle (HSC) and the horizontal long axis of the condyle (HLC) in a subject from the group “Bilateral grinding” (**B**) Measurement of HSC and HLC in a patient from the group “Non-bilateral grinding” (**C**) Distribution of HSC of chewing pattern group “Bilateral grinding” and group “Non-bilateral grinding”, *: *p* < 0.05. (**D**) Measurement of the sagittal depth of condyle (SDC) in a patient from the group “Bilateral grinding” (**E**) Measurement of SDC in a patient from the group “Non-bilateral grinding” (**F**) Distribution of SDC of chewing pattern group “Bilateral grinding” and group “Non-bilateral grinding”. Chewing pattern group “Bilateral grinding”: grinding patterns on both sides. Chewing pattern group “Non-bilateral grinding”: with chopping, reversed, or crossover patterns on at least one side. This figure illustrates the relationship between the anteroposterior aspects of the mandibular condyle (HSC and SDC) and chewing patterns when compared to the mediolateral aspect (HLC). The chewing pattern group “Non-bilateral grinding” had a notably rounder mandibular condyle, particularly in the horizontal slice, whereas the chewing pattern group “Bilateral grinding” had a flatter shape in the anteroposterior aspect.

**Table 1 diagnostics-13-02177-t001:** The descriptive statistics of the variables.

Variables	Mean ± SD	Minimum	Maximum	*n* (%)
Dependent variables (TMJ morphology)				
Horizontal plane				
Long axis of condyle (HLC)	18.8 ± 3.4	11.0	26.4	
Short axis of condyle (HSC)	8.1 ± 1.2	5.2	11.4	
Horizontal condylar angle (HCA)	23.7 ± 11.0	3.2	59.1	
Coronal plane				
Long axis of condyle (CLC)	19.0 ± 3.3	12.3	27.2	
Radius of condyle at 45° (C45)	8.0 ± 1.6	4.3	13.4	
Radius of condyle at 90° (C90)	6.7 ± 1.6	2.1	10.2	
Radius of condyle at 135° (C135)	7.6 ± 1.6	3.0	10.6	
Joint space at 45° (CJS45)	3.1 ± 1.2	1.1	7.2	
Joint space at 90° (CJS90)	2.4 ± 0.8	1.1	5.4	
Joint space at 135° (CJS135)	2.4 ± 1.0	0.6	5.4	
Sagittal plane				
Height of condyle (SHC)	6.1 ± 1.5	1.8	9.8	
Depth of condyle (SDC)	7.8 ± 1.4	3.8	10.8	
Articular eminence inclination (SAEI)	32.6 ± 9.8	12.2	63.1	
Anterior joint space (SAJS)	2.0 ± 0.7	0.5	4.9	
Posterior joint space (SPJS)	2.1 ± 0.9	0.9	5.8	
Independent variables				
Chewing patterns				
Group “Bilateral grinding”	0			25 (24.5)
Group “Non-bilateral grinding”	1			77 (75.5)
SNB (°)	78.1 ± 5.3	68.1	92.2	
FMA (°)	31.0 ± 7.9	14	63.2	
Mandibular shift (mm)	2.3 ± 3.0	0	12.0	
Sex				
Male	0			29 (28.4)
Female	1			73 (71.6)
Age (yo)	24.1 ± 5.4			
TMD symptoms				
Absent	0			46 (45.1)
Present	1			56 (54.9)

“Bilateral grinding” denotes grinding patterns on both sides, while “Non-bilateral grinding” denotes chopping, reversed, or crossover patterns on at least one side. Abbreviations: SNB, the SNB angle; FMA, the Frankfort-mandibular plane angle; TMD, temporomandibular disorders; SD, standard deviation.

**Table 2 diagnostics-13-02177-t002:** The intra- and interobserver reliability of the TMJ variables.

TMJ Measurement Items	Intraobserver Reliability (*n* = 102)		Interobserver Reliability (*n* = 30)			
	Probability		Probability			
			Right TMJ		Left TMJ	
Long axis of condyle (HLC)	0.094	ns	0.071	ns	0.873	ns
Short axis of condyle (HSC)	0.065	ns	0.071	ns	0.071	ns
Horizontal condylar angle (HCA)	0.938	ns	0.804	ns	0.967	ns
Long axis of condyle (CLC)	0.671	ns	0.206	ns	0.105	ns
Radius of condyle at 45° (C45)	0.096	ns	0.059	ns	0.118	ns
Radius of condyle at 90° (C90)	0.109	ns	0.088	ns	0.142	ns
Radius of condyle at 135° (C135)	0.573	ns	0.103	ns	0.118	ns
Joint space at 45° (CJS45)	0.070	ns	0.070	ns	0.067	ns
Joint space at 90° (CJS90)	0.193	ns	0.375	ns	0.056	ns
Joint space at 135° (CJS135)	0.677	ns	0.071	ns	0.073	ns
Height of condyle (SHC)	0.118	ns	0.152	ns	0.083	ns
Depth of condyle (SDC)	0.661	ns	0.062	ns	0.059	ns
Articular eminence inclination (SAEI)	0.777	ns	0.616	ns	0.085	ns
Anterior joint space (SAJS)	0.179	ns	0.118	ns	0.153	ns
Posterior joint space (SPJS)	0.915	ns	0.407	ns	0.118	ns

Abbreviation: TMJ, Temporomandibular joint.

**Table 3 diagnostics-13-02177-t003:** Univariate analysis.

Dependent Variables (TMJ Morphology)	Probability	
Horizontal plane		
Long axis of condyle (HLC)	0.2556	
Short axis of condyle (HSC)	0.0060	*p* < 0.2
Horizontal condylar angle (HCA)	0.0521	*p* < 0.2
Coronal plane		
Long axis of condyle (CLC)	0.4254	
Radius of condyle at 45° (C45)	0.3519	
Radius of condyle at 90° (C90)	0.0504	*p* < 0.2
Radius of condyle at 135° (C135)	0.0061	*p* < 0.2
Joint space at 45° (CJS45)	0.4363	
Joint space at 90° (CJS90)	0.8465	
Joint space at 135° (CJS135)	0.5753	
Sagittal plane		
Height of condyle (SHC)	0.0793	*p* < 0.2
Depth of condyle (SDC)	0.0313	*p* < 0.2
Articular eminence inclination (SAEI)	0.8423	
Anterior joint space (SAJS)	0.2858	
Posterior joint space (SPJS)	0.4226	

**Table 4 diagnostics-13-02177-t004:** Multiple linear regression models.

Multiple Linear Regression Model with HSC as the Dependent Variable				
HSC	Coefficient	95% CI		Probability
Chewing patterns	0.612	0.163	1.061	0.008 **
SNB	0.041	−0.002	0.085	0.063
FMA	−0.05	−0.079	−0.021	0.001 **
Mandibular shift	−0.099	−0.166	−0.033	0.004 **
Sex	−0.527	−0.968	−0.087	0.020 *
Age	0.037	0.002	0.073	0.040 *
TMD symptoms	−0.329	−0.716	0.059	0.095
Multiple linear regression model with HCA as the dependent variable				
HCA	Coefficient	95% CI		Probability
Chewing patterns	−2.532	−6.966	1.902	0.260
SNB	−0.835	−1.267	−0.403	<0.001 ***
FMA	0.239	−0.050	0.527	0.103
Mandibular shift	0.019	−0.638	0.676	0.954
Sex	0.490	−3.862	4.842	0.824
Age	0.238	−0.115	0.590	0.184
TMD symptoms	−1.731	−5.555	2.093	0.371
Multiple linear regression model with C90 as the dependent variable				
C90	Coefficient	95% CI		Probability
Chewing patterns	0.412	−0.234	1.059	0.208
SNB	0.080	0.017	0.142	0.014 *
FMA	−0.038	−0.080	0.004	0.079
Mandibular shift	−0.055	−0.151	0.041	0.259
Sex	−0.814	−1.448	−0.180	0.012 *
Age	−0.004	−0.056	0.004	0.865
TMD symptoms	0.273	−0.285	0.830	0.334
Multiple linear regression model with C135 as the dependent variable				
C135	Coefficient	95% CI		Probability
Chewing patterns	0.689	0.101	1.277	0.022 *
SNB	0.093	0.036	0.150	0.002 **
FMA	−0.051	−0.089	−0.013	0.009 **
Mandibular shift	−0.080	−0.167	0.007	0.071
Sex	−0.818	−1.395	−0.241	0.006 **
Age	−0.011	−0.057	0.036	0.656
TMD symptoms	0.105	−0.402	0.613	0.680
Multiple linear regression model with SHC as the dependent variable				
SHC	Coefficient	95% CI		Probability
Chewing patterns	0.357	−0.268	0.982	0.260
SNB	0.063	0.002	0.124	0.044 *
FMA	−0.043	−0.084	−0.003	0.036 *
Mandibular shift	−0.037	−0.130	0.055	0.426
Sex	−0.687	−1.300	−0.073	0.029 *
Age	0.057	0.007	0.107	0.025 *
TMD symptoms	−0.362	−0.901	0.178	0.186
Multiple linear regression model with SDC as the dependent variable				
SDC	Coefficient	95% CI		Probability
Chewing patterns	0.551	−0.038	1.141	0.066
SNB	0.036	−0.021	0.094	0.212
FMA	−0.054	−0.092	−0.016	0.006 **
Mandibular shift	−0.08	−0.167	0.007	0.072
Sex	−0.514	−1.093	0.064	0.081
Age	0.016	−0.031	0.063	0.504
TMD symptoms	−0.401	−0.909	0.107	0.120

Abbreviations: HSC, horizontal short axis of condyle; HCA, horizontal condylar angle; C90, radius of condyle at 90°; C135, radius of condyle at 135°; SHC, sagittal height of condyle; SDC, sagittal depth of condyle; 95%CI, 95% confidence interval; SNB, the SNB angle; FMA, the Frankfort-mandibular plane angle; TMD, temporomandibular disorders. *: *p* < 0.05; **: *p* < 0.01; ***: *p* < 0.001.

**Table 5 diagnostics-13-02177-t005:** Comparison of TMJ measurements between two groups of chewing patterns.

TMJ Morphology	Chewing Patterns		Probability
	“Bilateral Grinding” (*n* = 25)	“Non-Bilateral Grinding” (*n* = 77)	
HSC (mm)	7.5 ± 0.8	8.3 ± 1.3	0.008 **
HCA (°)	27.4 ± 13.9	22.5 ± 9.7	0.260
C90 (mm)	6.1 ± 1.8	6.8 ± 1.5	0.208
C135 (mm)	6.8 ± 1.7	7.8 ± 1.5	0.022 *
SDC (mm)	7.3 ± 1.3	8.0 ± 1.4	0.066

Means ± standard deviations are shown. “Bilateral grinding” denotes grinding patterns on both sides, while “Non-bilateral grinding” denotes chopping, reversed, or crossover patterns on at least one side. Abbreviations: HSC, horizontal short axis of condyle; HCA, horizontal condylar angle; C90, radius of condyle at 90°; C135, radius of condyle at 135°; SHC, sagittal height of condyle; SDC, sagittal depth of condyle. *: *p* < 0.05; **: *p* < 0.01.

## Data Availability

The data that support the findings of this study are available upon request from the corresponding author. The data are not publicly available due to privacy and ethical restrictions.

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
