# Peer review of "Association between the Temporomandibular Joint Morphology and Chewing Pattern"

_diagnostics, 2023, doi:10.3390/diagnostics13132177_

Round 1

Reviewer 1 Report

Dear authors,

The papers was interesting and apparently novel.

The metric technique designed to assess the morphology of the TMJ was well described, but I feel that the paper would benefit from a structure more EQUATOR-like. Please, check the EQUATOR structure (probably the STROBE one for cross-sectional studies) and adapt.

I missed more recent studies (e.g. 2020-2023) on the topic.

I missed a proper discussion about the intra- and inter-observer rates. How do you explain the variation between the reproducibility of the parameters studied.

How do you explain the application of this study in practice. By knowing the morphology of the condyle what benefit do we have to treat a patient?

Can we reset the chewing pattern of a patient do avoid degenerative morphological changes of the condyle?

Author Response

Responses to Reviewer #1

We truly appreciate the time and effort that Reviewer #1 have dedicated to providing the valuable feedback on the article. We are grateful to the insightful comments. We prepared a point-by-point response as follows, and revised our manuscript according to the comments.

Comment 1: The metric technique designed to assess the morphology of the TMJ was well described, but I feel that the paper would benefit from a structure more EQUATOR-like. Please, check the EQUATOR structure (probably the STROBE one for cross-sectional studies) and adapt.

Response: We have conducted a thorough examination of the STROBE checklist for cross-sectional studies, and we are confident that our manuscript adheres to the checklist with some necessary adaptations. To provide further clarification, we have included the sample size information for TMJ in the results section. These modifications can be found on page 6, paragraph 2, lines 1-4:

“Based on the criteria we established for selecting TMJ sides, a total of 58 TMJs located on the deviated side were chosen from patients exhibiting mandibular shift. Additionally, 22 TMJs were selected from the left side and 22 from the right side of individuals without mandibular shift. In total, the study utilized 102 TMJs to gather data and conduct analyses.”

Comment 2. I missed more recent studies (e.g., 2020-2023) on the topic.

Response: Accordingly, we have added 2 recent studies as references:

- page 12, paragraph 2, line 4

  1. Ma, H.; Shu, J.; Zheng, T.; Liu, Y.; Shao, B.; Liu, Z. The effect of mandibular movement on temporomandibular joint morphology while eating French fries. Ann Anat 2022, 244, 151992, doi:10.1016/j.aanat.2022.151992.

- page 13, paragraph 2, line 3

  1. Rustia, S.; Lam, J.; Tahir, P.; Kharafi, L.A.; Oberoi, S.; Ganguly, R. Three-dimensional morphologic changes in the temporomandibular joint in asymptomatic patients who undergo orthodontic treatment: A systematic review. Oral Surg Oral Med Oral Pathol Oral Radiol 2022, 134, 397-406, doi:10.1016/j.oooo.2022.05.003.

Comment 3. I missed a proper discussion about the intra- and inter-observer rates. How do you explain the variation between the reproducibility of the parameters studied.

Response: We agree with the suggestion and have incorporated it throughout the manuscript. Specifically, we have included a comprehensive explanation of the intra- and inter-observer reliabilities of the TMJ measurements by adding Table2. Relevant discussion can be found on page 12, paragraph 4:

“Table 2 presents the results of the analysis conducted to assess the intra- and inter-observer reliabilities of the TMJ measurements. The findings reveal that there were no statistically significant differences observed between the levels of reliability for both intra- and inter-observer assessments, as indicated by p-values greater than 0.05. This implies that the agreement or consistency among observers and the consistency over time for the same observer were comparable for all TMJ measurements evaluated in the study. There-fore, the data in Table 2 suggest that the TMJ variables examined exhibited similar levels of reliability regardless of whether they were assessed by the same observer on multiple occasions or by different observers independently.”

Comment 4. How do you explain the application of this study in practice. By knowing the morphology of the condyle what benefit do we have to treat a patient?

Response: Our study revealed a significant correlation between TMJ morphology and chewing patterns, even after considering other covariates in multiple linear regression models. However, further research is necessary to elucidate the underlying mechanisms behind this correlation. Prospective studies with larger sample sizes and improved selection criteria for the study population are needed to establish a cause-and-effect relationship between these correlations. Knowing the morphology of the mandibular condyle can be an index for diagnosing various diseases, and it may be a guideline for improving habits and various dental treatments.

Comment 5. Can we reset the chewing pattern of a patient do avoid degenerative morphological changes of the condyle?

Response: In order to change an undesired chewing pattern, the cause should first be considered. Depending on the cause, improvement of habits such as mastication training, various occlusal improvement treatments, and surgical orthodontic treatment are considered. Additional investigations are needed to understand the mechanisms and long-term effects of resetting chewing patterns for preventing degenerative condylar changes.

Reviewer 2 Report

Dear authors,

this is an interesting cross sectional retrospective study, correlating chewing patters and tmj morphology. I read with great interest the methodology and results of your manuscript, and I have a few comments that needs to be addressed.

1. As this is a study based on patients seeking orthodontic treatment, could this be assessed as a confounding factor in your study? You do address this point, as these are not results from a general population pool, but it needs further clarification as this is a statistical bias on the conclusion of this manuscript.

2. As the methodology of the coronal process was highly specified, why was there a need for a second observer for reliability?

3. How do you address skeletal discrepancies, as well as correlations between chewing patters and age, sex?

The final paragraph of the discussion as well as the conclusion needs some clarification on how the results of this study correlate with the effects of orthodontic therapy on TMJ morphology. This is not directly provided by the presented results.

In general, it is a well-designed and rather interesting study, and I recommend in favour of publication, after addressing these minor issues.

Quality of english language is good, minor refinements required.

Author Response

Responses to Reviewer #2

We truly appreciate the time and effort that Reviewer #2 have dedicated to provide the valuable feedback on the article. We are grateful to the insightful comments. We prepared a point-by-point response as follows, and revised our manuscript according to the comments.

Comment 1: As this is a study based on patients seeking orthodontic treatment, could this be assessed as a confounding factor in your study? You do address this point, as these are not results from a general population pool, but it needs further clarification as this is a statistical bias on the conclusion of this manuscript.

Response: We appreciate the valuable inquiry. In addition to addressing the potential confounding factor related to the study population seeking orthodontic treatment, it is important to highlight that the main objective of our study is to investigate the relationship between TMJ morphology and chewing patterns. While the study population may introduce a statistical bias, our focus on examining this specific relationship allows for a more targeted analysis of the variables of interest. By emphasizing this objective, we aim to provide further clarification regarding the scope and purpose of the study and its implications for understanding the association between TMJ morphology and chewing patterns within the context of orthodontic treatment-seeking individuals.

Comment 2. As the methodology of the coronal process was highly specified, why was there a need for a second observer for reliability?

Response: We greatly appreciate your insightful comment. Even the methodology is highly specified, the involvement of the second observer for reliability assessment serves several important purposes. By doing that we are able to validate the methodology, minimize the bias, estimate the generalizability, ensure the quality assurance, and enhance the scientific rigor. This additional layer of objectiveness strengthens the overall reliability and validity of the study.

Comment 3. How do you address skeletal discrepancies, as well as correlations between chewing patters and age, sex?

Response: We appreciate the reviewer's question regarding the handling of skeletal discrepancies and correlations between chewing patterns, age, and sex in our study. To address skeletal discrepancies, we included several parameters of skeletal morphology including the SNB angle, FMA, and mandibular shift, as covariates in the regression models. The SNB angle, evaluated using tracings of the lateral cephalometric radiographs, assesses the antero-posterior position of the mandible relative to the upper cranial structures. Additionally, the FMA was assessed to evaluate the relationship between the Frankfort horizontal plane and the mandibular plane. We also assessed the lateral mandibular shift using the postero-anterior cephalometric radiographs. Regarding the correlations between the chewing patterns with age and sex, we accounted for these factors by including them as covariates in our analysis. This approach allowed us to examine the specific associations between chewing patterns and TMJ morphology while considering the potential influences of age and sex.

Comment 4. The final paragraph of the discussion as well as the conclusion needs some clarification on how the results of this study correlate with the effects of orthodontic therapy on TMJ morphology. This is not directly provided by the presented results.

Response: We acknowledge the reviewer's comment regarding the need for clarification on how the results of our study correlate with the effects of orthodontic therapy on TMJ morphology, particularly in the final paragraph of the discussion and the conclusion. In the meantime, while our study focused on investigating the association between TMJ morphology and bilateral chewing patterns, we admit that the direct effects of orthodontic therapy on TMJ morphology were not within the scope of this study. Findings of this study suggest that the morphology of the mandibular condyle may be associated with not only skeletal morphology but also masticatory function, such as chewing patterns that are related to occlusion and orthodontic treatment.

Reviewer 3 Report

This study could be very interesting for clinical practice. 

Some points should be improves:

1. Abstract: "In multiple linear regression models adjusted for other clvsriates, the length of the horizontal short axis of the condyle... were significantly associated with chewing patterns". Please add also how this associatuon id, with which kind of pattern... etc.

2. Introduction: Explain what FMA abbreviation is.

3. Results:

I still dont understand why there are less TMJs than patients, if we want to associate chewing wirh morphology your are still losing almost 60 TMJ to associate with the chewing morphology, that means 50% of the total sample. I need a good justification or remove those patients whose TMJ both were no availabñe in statistical analysis. 

Author Response

Responses to Reviewer #3

We truly appreciate the time and effort that Reviewer #3 have dedicated to providing the valuable feedback on the article. We are grateful to the insightful comments. We prepared a point-by-point response as follows, and revised our manuscript according to the comments.

Comment 1. Abstract: "In multiple linear regression models adjusted for other covariates, the length of the horizontal short axis of the condyle... were significantly associated with chewing patterns". Please add also how this association id, with which kind of pattern... etc.

Response: We have revised the last two sentences of the Abstract on page 1 within the word limit according to the reviewer comments:

We have made the necessary revisions in Abstract accordingly:

“In multiple linear regression models adjusted for other covariates, the length of the horizontal short axis of the condyle and radius of the condyle at 135° from the medial pole were significantly (p<0.05) associated with the chewing patterns in the frontal plane on the working side. "Non-bilateral grinding" displayed a more rounded shape of the mandibular condyle. Conversely, “bilateral grinding” exhibited a flatter shape in the anteroposterior aspect. These findings suggest that the mandibular condyle morphology might be related to skeletal and masticatory function, including chewing patterns.”

Comment 2. Introduction: Explain what FMA abbreviation is.

Response: We have made the necessary adjustments. The specific changes can be located on page 1, paragraph 2, line 3:

“Patients with a larger Frankfort-mandibular plane angle (FMA) had a smaller linear measurement of the mandibular condyle size, although the difference was not statistically significant [5,9].”

Comment 3. Results: I still don’t understand why there are less TMJs than patients, if we want to associate chewing with morphology you are still losing almost 60 TMJ to associate with the chewing morphology, that means 50% of the total sample. I need a good justification or remove those patients whose TMJ both were no available in statistical analysis. 

Response: We revised the second paragraph of the Discussion on page 11 as to why not all temporomandibular joints of patients were studied in this study:

”Preliminary studies revealed that most cases of large lateral mandibular deviation in our sample were skeletal Class III cases, and that the condylar morphology of the non-deviated sides was large due to overgrowth. Therefore, we thought that including the mandibular condyles on the non-deviated sides would mask the morphological characteristics of the mandibular condyles on the deviated sides. One of the purposes of this study was to clarify the relationship between mandibular condyle morphology and gum chewing patterns on the deviated sides. Therefore, we decided not to include the mandibular condyles on the non-deviated sides in cases where mandibular deviation was evident.”
